# Carpal tunnel syndrome in dentists compared to other populations: A nationwide population-based study in Taiwan

Wei-Ta Huang[1]☯, Chia-Ti Wang[1]☯, Chung-Han Ho[2,3], Yi-Chen Chen[2], Yu-Chieh Ho[1], Chien-Chin Hsu[1], Hung-Jung Lin[1,4], Jhi-Joung Wang[5,6], Lian-Ping Mau[7,8]*, Chien-Cheng Huang [1,9,10,11]*

1 Department of Emergency Medicine, Chi Mei Medical Center, Tainan, Taiwan, 2 Department of Medical Research, Chi Mei Medical Center, Tainan, Taiwan, 3 Department of Information Management, Southern Taiwan University of Science and Technology, Tainan, Taiwan, 4 Department of Emergency Medicine, Taipei Medical University, Taipei, Taiwan, 5 Department of Anesthesiology, Chi Mei Medical Center, Tainan, Taiwan, 6 Department of Anesthesiology, National Defense Medical Center, Taipei, Taiwan, 7 Department of Periodontics, Chi Mei Hospital, Chiali, Tainan, Taiwan, 8 Department of Senior Welfare and Services, Southern Taiwan University of Science and Technology, Tainan, Taiwan, 9 Department of Emergency Medicine, Kaohsiung Medical University, Kaohsiung, Taiwan, 10 Department of Environmental and Occupational Health, College of Medicine, National Cheng Kung University, Tainan, Taiwan, 11 School of Medicine, College of Medicine, National Sun Yat-sen university, Kaohsiung, Taiwan

☯ These authors contributed equally to this work.
* chienchenghuang@yahoo.com.tw (CCH); lianpingmau@yahoo.com.tw (LPM)

**Data Availability Statement:** Data are available from the NHIRD published by Taiwan National Health Insurance Bureau. Due to legal restrictions

## Abstract

### Background

Dentists may be at a higher risk of developing carpal tunnel syndrome (CTS) because of their use of frequent wrist and vibratory instruments at work; however, this issue remains unclear. Therefore, we conducted this study to clarify it.

### Methods

Taiwan National Health Insurance Research Database was used for this nationwide population-based study. We identified 11,084 dentists, 74,901 non-dentist healthcare professionals (HCPs), and identical number of age- and sex-matched participants from the general population. Participants who had the diagnosis of CTS before 2007 were excluded. Between 2007 and 2011, the risk of developing CTS among dentists, non-dentist HCPs, and the general population was compared by following their medical histories.

### Results

The cumulative incidence rate of CTS among dentists was 0.5% during the 5-year follow-up period. In dentists, the risk was higher in women (women: 0.7%; men: 0.4%) and older individuals ($\geq$60 years: 1.0%; <60 years: 0.4%). After adjusting for age, sex, and underlying comorbidities, dentists had a lower risk of CTS than the general population (adjusted odds ratio [AOR]: 0.65, 95% confidence interval [CI]: 0.45–0.92). Dentists had a higher risk for CTS compared with non-dentist HCPs, although the difference was not statistically significant (AOR: 1.21; 95% CI: 0.90–1.64).

imposed by the government of Taiwan in relation to the "Personal Information Protection Act," data cannot be made publicly available. Requests for data can be sent as a formal proposal to the NHIRD (http://nhird.nhri.org.tw).

**Funding:** This study was supported by the Grant CMFHR 112072 and Grant Physician-Scientist 11001 from the Chi Mei Medical Center. No additional external funding was received for this study. The funders had no role in study design, data collection and analysis, decision to publish, or preparation of the manuscript.

**Competing interests:** The authors have declared that no competing interests exist.

## Conclusions

In CTS, dentists had a lower risk than the general population and a trend of higher risk than non-dentist HCPs. The difference between dentists and non-dentist HCPs suggests that we should pay attention to dentists for potential occupational risk of this disease. However, further studies are warranted to better clarify it.

## Introduction

Compression of the median nerve at the wrist is the cause of carpal tunnel syndrome (CTS), which is the most prevalent peripheral mononeuropathy [1, 2]. The prevalence of CTS is estimated to be 2.7% to 5.8% of the general population, and the mean annual crude incidence was about 329 cases per 100,000 person-years [2]. CTS leads to pain, numbness, and tingling in the upper extremities, which may contribute to work disability and increased economic burden [3]. Common etiologies of CTS are repetitive maneuvers of hand and wrist, use of handheld powered vibratory tools, obesity, pregnancy, arthritis, hypothyroidism, diabetes, trauma, mass lesions, amyloidosis, sarcoidosis, multiple myeloma, and leukemia [2, 4].

Previous studies reported that CTS is associated with some occupations, including the jobs entailing the use of vibratory tools, assembly work, and food processing and packing [4]. Dentists have several risk factors for CTS, including repetitive flexion and extension of hand and wrist, sustained grasps to sharp edges on instrument handles, forceful work, and extended use of vibratory instruments [5]. A cross-sectional study in the United States in 2001 reported that 2.9% of dentists were diagnosed with CTS [6]. A study in Iran in 2012 reported that 16.7% of dentists had CTS, and the prevalence increased with age [7]. Another study in Saudi Arabia reported that the prevalence rate of CTS was 30% among dentists in the country [8]. According to the studies above, the prevalence of dentists varied greatly according to different nations and diagnosis methods. There was no study about CTS of dentists in Asian population. In addition, the comparison of CTS between dentists and the general population and between dentists and non-dentist healthcare professionals (HCPs) remains unclear. Therefore, we conducted this study to provide clarity on these issues.

## Materials and methods

### Data source

This nationwide population-based study was conducted using the Taiwan National Health Insurance Research Database (NHIRD), which comes from the National Health Insurance program that includes approximately the entire Taiwan population [9]. The NHIRD provides epidemiological studies with a real-world population-based database and opportunities to solve questions that are difficult to answer using randomized control trial [9]. The accuracy and contribution of NHIRD-based health care studies have been proven in numerous studies [9]. Authors do not possess any information that could identify individual participants either during or after the data collection process.

### Study design, setting, and participants

Participants in the NHIRD who were diagnosed with CTS before 2007 were excluded (Fig 1). The criteria for diagnosing CTS were based on a diagnosis using ICD-9-CM code of 354.0 during at least one hospitalization or outpatient clinic visit. We did not use any instruments to collect data about the participants. In Taiwan, the diagnosis of CTS is typically based on clinical

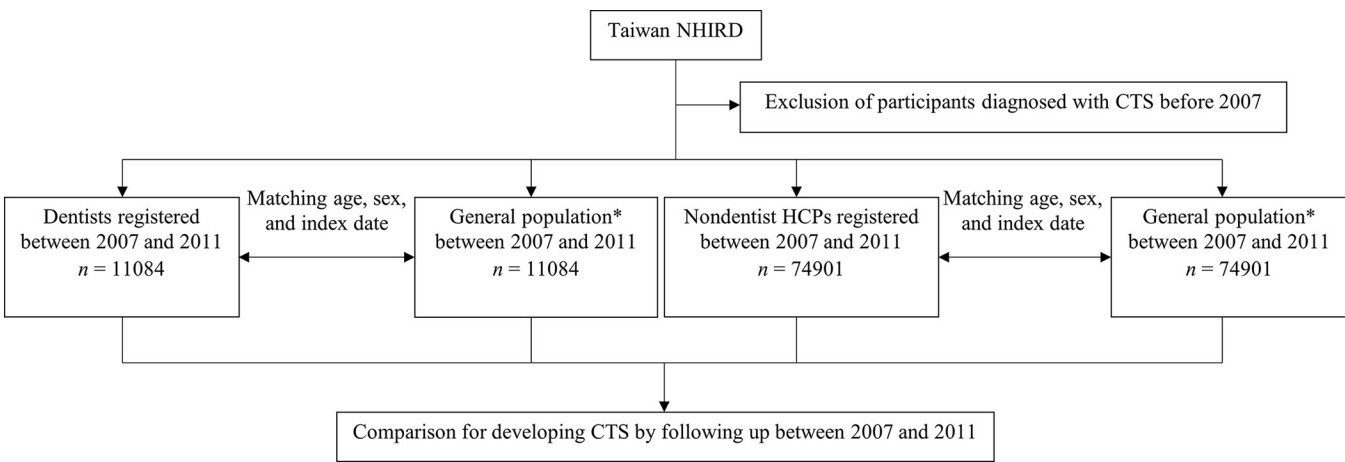

**Fig 1. Flowchart of this study.** NHIRD, National Health Insurance Research Database; CTS, carpal tunnel syndrome; HCP, healthcare professionals. *Non-HCP comparisons.

symptoms such as numbness, tingling, and pain in the hand, and is usually confirmed through electrodiagnostic testing by a qualified physician [10]. Dentists who were registered between 2007 and 2011 were identified. We identified participants who were not HCP by matching age, sex, and index date at 1:1 ratio as the non-HCPs comparisons (i.e., general population). The analyses included non-dentist HCPs who were registered between 2007 and 2011, as well as an identical number of non-HCP comparisons. The index date is the day on which the participant visited an outpatient clinic or was admitted to the hospital [11, 12]. To help identify the comparisons, we used index date. Physicians, pharmacists, medical technicians, clinical experts, consultant experts, audiologists, language experts, dietitians, and social workers were among the non-dentist HCPs [11, 12]. We included non-dentist HCPs in this study because they have a similar working environment and socioeconomic status with dentists and may help us investigate the difference with dentists. The participants were divided into three age subgroups: ≤34, 35–59, and ≥60 years old [11, 12]. Underlying comorbidities were defined as diabetes (ICD-9-CM code 250), hypertension (ICD-9-CM code: 401–405), hyperlipidemia (ICD-9-CM code: 272), malignancy (ICD-9-CM code: 140–208), stroke (ICD-9-CM code: 436–438), coronary artery disease (CAD) (ICD-9-CM code: 410–414), chronic obstructive pulmonary disease (COPD) (ICD-9-CM code: 496), liver disease (ICD-9-CM code: 570–576), renal disease (ICD-9-CM code: 580–593), and mental disorder (ICD-9-CM code: 290–319) [11, 12]. The underlying comorbidities included in this study were determined based on the diagnosis of these diseases during at least one hospitalization or at least three times outpatient care [11, 12].

## Comparison of the risk of developing CTS

By following up on their medical histories from 2007 to 2011, the risk of developing CTS was compared between dentists and the general population, between non-dentist HCPs and the general population, and between dentists and non-dentist HCPs. Stratified analyses for age and sex subgroups were also performed to assess whether age and sex were effect modifiers.

## Ethics statement

This study was strictly conducted according to the Declaration of Helsinki and approved by the Institutional Review Board at Chi Mei Medical Center. Since the NHIRD contains de-

identified information, informed consents from the participants are waived, and the rights and welfare of the participants are not affected.

## Statistical analysis

In the comparison of age, sex, and underlying comorbidities between dentists and the general population, between non-dentist HCPs and the general population, and between dentists and non-dentist HCPs, we employed independent *t*-test for continuous variables and chi-square test for categorical variables. To compare the risk of developing CTS between dentists and the general population, as well as between non-dentist HCPs and the general population, we performed conditional logistic regression analyses. These analyses were adjusted for various confounding factors such as diabetes, hypertension, hyperlipidemia, malignancy, stroke, CAD, COPD, liver disease, renal disease, and mental disorder. The risk of developing CTS between dentists and non-dentist HCPs was evaluated using unconditional logistic regression analysis by adjusting for age, sex, diabetes, hypertension, hyperlipidemia, malignancy, stroke, CAD, COPD, liver disease, renal disease, and mental disorder. All analyses were performed using SAS 9.4 for Windows (SAS Institute, Cary, NC, USA), with a significance level of 0.05 (two-tails).

## Results

In total, we identified 11,084 dentists and 11,084 age- and sex-matched participants from the general population for comparison (Table 1). The mean age (±standard deviation) in dentists

**Table 1.  Comparisons of age, sex, underlying comorbidities, and living areas among dentists, non-dentist HCPs, and the general population (non-HCP comparisons).**

| | Dentists (*n* = 11,084) | General population (*n* = 11,084) | *p*-value | Non-dentist HCPs (*n* = 74901) | General population (*n* = 74901) | *p*-value |
|---|---|---|---|---|---|---|
| Age (years) | 43.7 ± 11.3 | 43.7 ± 11.3 | >0.999 | 42.6 ± 12.2 | 42.6 ± 12.2 | >0.999 |
| Age (years) | | | | | | |
| ≤34 | 2,969 (26.8) | 2,969 (26.8) | >0.999 | 23,375 (31.2) | 23,375 (31.2) | >0.999 |
| 35−59 | 7,215 (65.1) | 7,215 (65.1) | | 45,120 (60.2) | 45,120 (60.2) | |
| ≥60 | 900 (8.1) | 900 (8.1) | | 6,406 (8.6) | 6,406 (8.6) | |
| Sex | | | | | | |
| Male | 8,415 (75.9) | 8,415 (75.9) | | 46,724 (62.4) | 46,724 (62.4) | |
| Female | 2,669 (24.1) | 2,669 (24.1) | >0.999 | 28,177 (37.6) | 28,177 (37.6) | >0.999 |
| Underlying comorbidity | | | | | | |
| Diabetes | 806 (7.3) | 883 (8.0) | 0.051 | 4,508 (6.0) | 5,158 (6.9) | **<0.001***|
| Hypertension | 1,976 (17.8) | 1,847 (16.7) | 0.022 | 12,392 (16.5) | 10,612 (14.2) | **<0.001***|
| Hyperlipidemia | 1,587 (14.3) | 1,242 (11.2) | <0.001 | 10,457 (14.0) | 7,214 (9.6) | **<0.001***|
| Malignancy | 246 (2.2) | 289 (2.6) | 0.060 | 1,952 (2.6) | 1,912 (2.6) | 0.514 |
| Stroke | 198 (1.8) | 295 (2.7) | <0.001 | 1,459 (2.0) | 2,050 (2.7) | **<0.001***|
| CAD | 586 (5.3) | 503 (4.5) | 0.010 | 3,345 (4.5) | 3,163 (4.2) | **0.021***|
| COPD | 490 (4.4) | 659 (6.0) | <0.001 | 5,340 (7.1) | 4,418 (5.9) | **<0.001***|
| Liver disease | 1,200 (10.8) | 1,114 (10.1) | 0.059 | 7,517 (10.0) | 6,649 (8.9) | **<0.001***|
| Renal disease | 127 (1.2) | 170 (1.5) | 0.012 | 915 (1.2) | 1,123 (1.5) | **<0.001***|
| Mental disorder | 933 (8.4) | 1,217 (11.0) | <0.001 | 9,278 (12.4) | 8,444 (11.3) | **<0.001***|

Data are number (%) or mean ± SD. HCP, healthcare professional; CAD, coronary artery disease; COPD, chronic obstructive pulmonary disease; SD, standard deviation. *There is a statistically significant difference with a p-value of less than 0.05.

and the general population was 43.7 ± 11.3 years. In terms of age subgroups, 26.8% were aged ≤34 years, 65.1% were aged 35–59 years, and 8.1% were aged ≥60 years. Most dentists were men (75.9%). Compared with the general population, dentists had higher underlying comorbidities of hypertension (17.8% vs. 16.7%), hyperlipidemia (14.3% vs. 11.2%), and CAD (5.3% vs. 4.5%), but lower stroke (1.79% vs. 2.66%), COPD (4.4% vs. 6.0%), renal disease (1.2% vs. 1.5%), and mental disorder (8.4% vs. 11.0%). For comparison, we identified identical numbers of age- and sex-matched participants from the general population, as well 74,901 non-dentist HCPs. The mean age (±standard deviation) of these participants was 42.6 ± 12.2 years. Regarding age subgroups, 31.2% of them were aged ≤34 years, 60.2% were aged 35–59 years, and 8.6% were aged ≥60 years. Male sex predominated in non-dentist HCPs (62.4%). Compared with the general population, non-dentists HCPs had higher underlying comorbidities of hypertension (16.5% vs. 14.2%), hyperlipidemia (14.0% vs. 9.6%), CAD (4.5% vs. 4.2%), COPD (7.1% vs. 5.9%), liver disease (10.0% vs. 8.9%), and mental disorder (12.4% vs. 11.3%), but lower diabetes (6.0% vs. 6.9%), stroke (2.0% vs. 2.7%), and renal disease (1.2% vs. 1.5%).

Between 2007 and 2011, the cumulative incidence rates of CTS in dentists and the general population were 0.5% and 0.7%, respectively (Table 2). Compared with the general population, dentists had a lower risk of developing CTS after adjusting for diabetes, hypertension, hyperlipidemia, malignancy, stroke, CAD, COPD, liver disease, renal disease, and mental disorder (adjusted odds ratio [AOR]: 0.65, 95% confidence interval [CI]: 0.45–0.92). The cumulative incidence rates of CTS in male and female dentists were 0.4% and 0.7%, respectively. In the dentists, CTS was higher with aging (0.4% in the age subgroups of ≤34 years and 35–59 years

**Table 2. Comparison of the risk for CTS between dentist and the general population (non-HCP comparisons) by conditional logistic regression.**

| Variable | Number (%) | OR (95% CI) | AOR (95% CI)† | p-value‡ |
|---|---|---|---|---|
| Dentists (n = 11084) | 50 (0.5) | 0.63 (0.44–0.90) | 0.65 (0.45–0.92) | **0.017*** |
| General population (n = 11,084) | 79 (0.7) | 1.0 (reference) | 1.0 (reference) | |
| Sex subgroup | | | | |
| Male | | | | |
| Dentists (n = 8,415) | 31 (0.4) | 0.61 (0.39–0.95) | 0.61 (0.39–0.97) | **0.036*** |
| General population (n = 8,415) | 51 (0.6) | 1.0 (reference) | 1.0 (reference) | |
| Female | | | | |
| Dentists (n = 2,669) | 19 (0.7) | 0.68 (0.38–1.22) | 0.68 (0.37–1.24) | 0.206 |
| General population (n = 2,669) | 28 (1.1) | 1.0 (reference) | 1.0 (reference) | |
| Age subgroup | | | | |
| ≤34 years | | | | |
| Dentists (n = 2,969) | 11 (0.4) | 1.10 (0.47–2.59) | 1.15 (0.48–2.74) | 0.760 |
| General population (n = 2,969) | 10 (0.3) | 1.0 (reference) | 1.0 (reference) | |
| 35–59 years | | | | |
| Dentists (n = 7,215) | 30 (0.4) | 0.52 (0.33–0.80) | 0.53 (0.34–0.82) | **0.005*** |
| General population (n = 7,215) | 58 (0.8) | 1.0 (reference) | 1.0 (reference) | |
| ≥60 years | | | | |
| Dentists (n = 900) | 9 (1.0) | 0.82 (0.34–1.97) | 0.89 (0.36–2.22) | 0.797 |
| General population (n = 900) | 11 (1.2) | 1.0 (reference) | 1.0 (reference) | |

CTS, carpal tunnel syndrome; HCP, health care professional; OR, odds ratio; AOR, adjusted odds ratio; CI, confidence interval.

*There is a statistically significant difference with a p-value of less than 0.05.

†Adjusted for diabetes, hypertension, hyperlipidemia, malignancy, stroke, coronary artery disease, chronic obstructive pulmonary disease, liver disease, renal disease, and mental disorder.

‡For AOR.

**Table 3. Comparison of the risk for CTS between non-dentist HCPs and the general population (non-HCP comparisons) by conditional logistic regression.**

| Variable | Number (%) | OR (95% CI) | AOR (95% CI)† | p-value‡ |
|---|---|---|---|---|
| Non-dentist HCPs (n = 74,901) | 322 (0.4) | 0.59 (0.52−0.68) | 0.56 (0.48−0.64) | **<0.001*** |
| General population (n = 74,901) | 542 (0.7) | 1.0 (reference) | 1.0 (reference) | |
| Sex subgroup | | | | |
| Male | | | | |
| Non-dentist HCPs (n = 46,724) | 171 (0.4) | 0.66 (0.54−0.80) | 0.61 (0.50−0.74) | **<0.001*** |
| General population (n = 46,724) | 259 (0.6) | 1.0 (reference) | 1.0 (reference) | |
| Female | | | | |
| Non-dentist HCPs (n = 28,177) | 151 (0.5) | 0.53 (0.44−0.65) | 0.52 (0.43−0.64) | **<0.001*** |
| General population (n = 28,177) | 283 (1.0) | 1.0 (reference) | 1.0 (reference) | |
| Age subgroup | | | | |
| ≤34 years | | | | |
| Non-dentist HCPs (n = 23375) | 49 (0.2) | 0.85 (0.58−1.24) | 0.80 (0.55−1.18) | 0.257 |
| General population (n = 23375) | 58 (0.3) | 1.0 (reference) | 1.0 (reference) | |
| 35−59 years | | | | |
| Non-dentist HCPs (n = 45120) | 229 (0.5) | 0.54 (0.46−0.64) | 0.50 (0.42−0.59) | **<0.001*** |
| General population (n = 45120) | 420 (0.9) | 1.0 (reference) | 1.0 (reference) | |
| ≥60 years | | | | |
| Non-dentist HCPs (n = 6406) | 44 (0.7) | 0.68 (0.46−1.01) | 0.75 (0.50−1.11) | 0.148 |
| General population (n = 6406) | 64 (1.0) | 1.0 (reference) | 1.0 (reference) | |

CTS, carpal tunnel syndrome; HCP, health care professional; OR, odds ratio; AOR, adjusted odds ratio; CI, confidence interval.

*There is a statistically significant difference with a p-value of less than 0.05.

†Adjusted for diabetes, hypertension, hyperlipidemia, malignancy, stroke, coronary artery disease, chronic obstructive pulmonary disease, liver disease, renal disease, and mental disorder.

‡For AOR.

vs. 1.0% in the age subgroup of ≥60 years). Stratified analyses showed that dentists in the male sex and age subgroup of 35−59 years had a lower risk of developing CTS than the general population (male sex, AOR: 0.61, 95% CI: 0.39−0.97; 35−59 years, AOR: 0.53, 95% CI: 0.34−0.82).

Non-dentist HCPs had a lower risk of developing CTS than the general population (AOR: 0.56, 95% CI: 0.48−0.64) (Table 3). Stratified analyses showed that non-dentist HCPs had a lower risk of developing CTS than the general population in both sex and age subgroup of 35−59 years.

Compared with non-dentist HCPs, the AOR of developing CTS in the dentists was 1.21; however, the difference was not statistically significant (95% CI: 0.90−1.64) (Table 4). The results were also similar in the stratified analyses according to sex and age subgroups.

## Discussion

This study showed that the cumulative incidence rate of CTS in dentists during the 5-year follow-up period was 0.5%. CTS was higher in female dentists and those who were older. Dentists had a lower risk of developing CTS than the general population, particularly in the male sex and age subgroup of 35−59 years. Non-dentist HCPs also had a lower risk of developing CTS than the general population. Compared with non-dentist HCPs, dentists had a trend of higher risk of CTS; however, the difference was not statistically significant.

This is the first study to reveal that dentists as well as non-dentist HCPs had a lower risk of CTS than the general population. The possible reason is that HCPs, including dentists and non-dentist HCPs have better medical knowledge and health concepts than the general

**Table 4. Comparison of the risk for CTS between dentists and non-dentist HCPs by unconditional logistic regression.**

| Variable | Number (%) | OR (95% CI) | AOR (95% CI)* | *p*-value† |
|---|---|---|---|---|
| Dentists (*n* = 11,084) | 50 (0.5) | 1.05 (0.78–1.42) | 1.21 (0.90–1.64) | 0.213 |
| Non-dentist HCPs (*n* = 74,901) | 322 (0.4) | 1.0 (reference) | 1.0 (reference) | |
| Sex subgroup | | | | |
| Male | | | | |
| Dentists (*n* = 8,415) | 31 (0.4) | 1.01 (0.69–1.48) | 1.11 (0.75–1.63) | 0.604 |
| Non-dentist HCPs (*n* = 46,724) | 171 (0.4) | 1.0 (reference) | 1.0 (reference) | |
| Female | | | | |
| Dentists (*n* = 2,669) | 19 (0.7) | 1.33 (0.83–2.15) | 1.48 (0.92–2.40) | 0.108 |
| Non-dentist HCPs (*n* = 28,177) | 151 (0.5) | 1.0 (reference) | 1.0 (reference) | |
| Age subgroup | | | | |
| ≤34 years | | | | |
| Dentists (*n* = 2,969) | 11 (0.4) | 1.77 (0.92–3.41) | 1.91 (0.99–3.69) | 0.055 |
| Non-dentist HCPs (*n* = 23,375) | 49 (0.2) | 1.0 (reference) | 1.0 (reference) | |
| 35−59 years | | | | |
| Dentists (*n* = 7,215) | 30 (0.4) | 0.82 (0.56–1.20) | 1.01 (0.69–1.49) | 0.950 |
| Non-dentist HCPs (*n* = 45,120) | 229 (0.5) | 1.0 (reference) | 1.0 (reference) | |
| ≥60 years | | | | |
| Dentists (*n* = 900) | 9 (1.0) | 1.46 (0.71–3.00) | 1.62 (0.78–3.36) | 0.194 |
| Non-dentist HCPs (*n* = 6,406) | 44 (0.7) | 1.0 (reference) | 1.0 (reference) | |

CTS, carpal tunnel syndrome; HCP, health care professional; OR, odds ratio; AOR, adjusted odds ratio; CI, confidence interval.

*Adjusted for diabetes, hypertension, hyperlipidemia, malignancy, stroke, coronary artery disease, chronic obstructive pulmonary disease, liver disease, renal disease, and mental disorder.

†For AOR.

population, which could help them reduce the risk of developing CTS. In the literature, there was no study comparing the risk of CTS between the dentists and the general population. In the general cognition, dentists are suspected to have higher risk for musculoskeletal diseases due to prolonged sitting and improper postures during work, frequent use of hand and wrist, and using vibratory tools [13, 14]. However, the results in this study indicate that this general cognition may not be correct. Dentists had a lower risk of developing lumbar herniated intervertebral disc than the general population (AOR: 0.80; 95% CI: 0.64–1.00), according to our study in 2019 using nationwide data [12]. Another study showed that there was no significant difference of risk for cervical herniated intervertebral between dentists and the general population (AOR: 1.2; 95% CI: 0.9–1.6) [11]. Another reason for the lower risk of CTS is that dentists may have self-treatment instead of seeking physician for diagnosis [15], which may underestimate the incidence of CTS.

This study showed that the cumulative incidence rate of CTS of 5 years in dentists was 0.5%, which was apparently lower than previous studies [5, 7, 16]. The possible explanation is that we used ICD-9-CM code of 354.0 in hospitalization or outpatient clinic for at least one time as the criteria of CTS, which is stricter than previous reports adopting questionnaire or interview [7, 8]. A nationwide population-based cohort study in Taiwan using the same criteria with us reported the <u>annual</u> incidence of CTS was 0.4% in the overall population [17], which was close to our data. The finding of higher risk of CTS in female and older dentists is compatible with previous reports [17]. The 5-year cumulative incidence rate of CTS in female dentists was 0.7%, higher than the 0.4% of male dentists. The possible pathophysiology is that women have a small wrist size and less elastic transverse carpal ligament [17–19]. In addition,

the lower cumulative incidence rate in this study may be due to a shorter period (5 years). Race may also explain the difference. As we mentioned above, the study in Taiwan reported a similar figure with us [17], which is different with other nations [6–8].

Although the difference between dentists and non-dentist HCPs was not significant, the trend of higher CTS still needs attention for the possible occupational risk for CTS in dentists. Dentists have an AOR of 1.91 in the age subgroup of ≤34 years compared with non-dentist HCPs. Dentists have more opportunities to perform vibratory instruments and frequent flexion and extension of wrist in daily work than other HCPs, which may increase the risk of CTS [4, 6–8, 13, 14, 20]. However, it needs further study, including more participants for clarifying this issue.

The total number of healthcare professionals in Taiwan in 2021 was 347,555, and the number of dentists was a subset of this [21]. For this study, we included all dentists in Taiwan, so the sample size was not a limitation. The non-dentist HCPs and general population were selected by matching the dentists, and the sample sizes were also large. According to the National Statistics, the total population in Taiwan in 2020 was approximately 23 million [22]. Given the large sample sizes in this study and the fact that we included all dentists in Taiwan, we believe that the study had adequate statistical power.

The major strength of this study is that it is the first to investigate the cumulative incidence of CTS in the dentists and comparison with other non-dentist HCPs and the general population using a nationwide population-based design. There are some limitations as the follows. First, while we adjusted for common underlying comorbidities in our study, including diabetes, hypertension, hyperlipidemia, stroke, CAD, COPD, liver disease, renal disease, and mental disorder, we acknowledge that certain associated factors were not included in our analysis, such as pregnancy status, obesity (e.g., body mass index), thyroid disorders, smoking status, alcohol intake, and certain athletic activities. Additionally, detailed information about working hours, years of entering the workforce, and previous trauma were not available, which may have confounded our results. These limitations should be taken into consideration when interpreting the findings of our study. Second, rather than seeking medical advice, dentists may treat themselves. As a result, using the physician's diagnosis in this study may underestimate the actual risk in dentists. Third, the follow-up period of 5 years may be too short to evaluate the effect of occupational exposure. Fourth, the workload of dentists may be different in different nations, and therefore, the results of this study may not be generalized to other nations. Further studies, including more detailed information about pregnancy status, obesity, thyroid disorders, smoking status, alcohol intake, and certain athletic activities, occupation, longer follow-up period, and validation in other nations, were warranted in the future.

## Conclusion

This is the first study to delineate the cumulative incidence of developing CTS in the dentists and compare it with non-dentist HCPs and the general population. Dentists, in contrast to the general cognition, had a lower risk of developing CTS than the general population. Better medical knowledge for prevention and self-treatment in dentists may play the role. Dentists had a trend of higher incidence of CTS than other HCPs, although the difference was not statistically significant, especially in the age subgroup of ≤34 years. The findings imply that dentists may have occupational risk for CTS than other HCPs, which requires more attention. These results could be useful for policy makers looking to promote occupational health among HCPs. However, additional studies are needed to investigate more detailed information about pregnancy status, obesity, thyroid disorders, smoking status, alcohol intake, certain athletic activities, and occupational exposure. These studies should have a longer follow-up period and should be validated in other countries.

## Supporting information

**S1 Checklist. STROBE statement—checklist of items that should be included in reports of observational studies.**
(DOCX)

## Acknowledgments

We thank Enago for the English revision.

## Author Contributions

**Conceptualization:** Wei-Ta Huang, Chia-Ti Wang, Chien-Cheng Huang.

**Data curation:** Chung-Han Ho, Yi-Chen Chen.

**Formal analysis:** Chung-Han Ho, Yi-Chen Chen.

**Funding acquisition:** Chien-Cheng Huang.

**Investigation:** Wei-Ta Huang, Chia-Ti Wang, Chien-Cheng Huang.

**Methodology:** Wei-Ta Huang, Chia-Ti Wang, Chien-Cheng Huang.

**Project administration:** Wei-Ta Huang, Chia-Ti Wang, Yu-Chieh Ho, Chien-Chin Hsu, Hung-Jung Lin, Jhi-Joung Wang, Lian-Ping Mau, Chien-Cheng Huang.

**Software:** Chung-Han Ho, Yi-Chen Chen.

**Supervision:** Chien-Cheng Huang.

**Validation:** Chien-Cheng Huang.

**Visualization:** Chien-Cheng Huang.

**Writing – original draft:** Wei-Ta Huang, Chia-Ti Wang, Chien-Cheng Huang.

**Writing – review & editing:** Wei-Ta Huang, Chia-Ti Wang, Chung-Han Ho, Yi-Chen Chen, Yu-Chieh Ho, Chien-Chin Hsu, Hung-Jung Lin, Jhi-Joung Wang, Lian-Ping Mau, Chien-Cheng Huang.

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
