## [Decision Letter · Decision Letter 0]

9 May 2023

PONE-D-23-10147Carpal tunnel syndrome in dentists compared to other populations: A nationwide population-based study in TaiwanPLOS ONE

Dear Dr. Huang,

Thank you for submitting your manuscript to PLOS ONE. After careful consideration, we feel that it has merit but does not fully meet PLOS ONE’s publication criteria as it currently stands. Therefore, we invite you to submit a revised version of the manuscript that addresses the points raised during the review process.

We look forward to receiving your revised manuscript.

Kind regards,

Hadi Ghasemi

Academic Editor

PLOS ONE

Journal Requirements:

- https://doi.org/10.1002/1348-9585.12036

In your revision ensure you cite all your sources (including your own works), and quote or rephrase any duplicated text outside the methods section. Further consideration is dependent on these concerns being addressed.

   "No."

   "No."

6. We note that you have indicated that data from this study are available upon request. PLOS only allows data to be available upon request if there are legal or ethical restrictions on sharing data publicly. For more information on unacceptable data access restrictions, please see http://journals.plos.org/plosone/s/data-availability#loc-unacceptable-data-access-restrictions. 

7. Your ethics statement should only appear in the Methods section of your manuscript. If your ethics statement is written in any section besides the Methods, please delete it from any other section. 

Reviewers' comments:

Reviewer's Responses to Questions

**Comments to the Author**

1. Is the manuscript technically sound, and do the data support the conclusions?

Reviewer #1: Yes

Reviewer #2: No

Reviewer #3: Yes

2. Has the statistical analysis been performed appropriately and rigorously? 

Reviewer #1: Yes

Reviewer #2: Yes

Reviewer #3: I Don't Know

3. Have the authors made all data underlying the findings in their manuscript fully available?

Reviewer #1: Yes

Reviewer #2: Yes

Reviewer #3: Yes

4. Is the manuscript presented in an intelligible fashion and written in standard English?

Reviewer #1: Yes

Reviewer #2: Yes

Reviewer #3: Yes

5. Review Comments to the Author

Reviewer #1: Dear authors,

Thank you for the interesting study, it is good manuscript, but I have very important comments to improve it more:

Methods:

1- regarding the study participants it is a study population survey in that case you need to discuss the power of study and number of participants compared to the whole population number in the conducted year.

2- regarding data collection all mentioned by authors was they referred to the data system only. Have the authors used an instrument to collect the needed data about participants, how it was developed and validated if so!?

3- How the authors have confirmed the diagnosis for carpal tunnel syndrome among participants as this is very important to discuss?

4- regarding the associated factors unfortunately the authors have missed very important associated factors for carpal tunnel syndrome which are: 1- pregnancy status for female participants as the pregnant female is more prone to develop carpal tunnel syndrome; 2- obesity (Body Mass Index(BMI)), as the obese person is more prone to develop carpal tunnel syndrome; 3- Thyroid disorder as hypothyroid patients are more prone to develop carpal tunnel syndrome; 4- Smoking status as smokers are more prone to develop carpal tunnel syndrome; 5- Alcohol intake status as alcohol consumers are more prone to develop carpal tunnel syndrome; 6- Any Rheumatoid diseases as rheumatoid patients are more prone to develop carpal tunnel syndrome; 7- Certain athletic activities such as tennis, those who play tennis are more prone to develop carpal tunnel syndrome.

Results:

1- Tables need reconstructing and keep lines between rows to show the results clearly.

2- The significant values should be marked by bold font or * and define that in the table footnotes

Wish you the best.

Reviewer #2: 1. In this study, the related diseases caused by some habits of dentists are described in detail.

2. The number of samples is very sufficient to reduce the error.

3. The idea and logic of this study are very clear, and I feel that the results are very convincing.

4. The statistical techniques used in this study are appropriate and persuasive.

Reviewer #3: Thank you very much for allowing me to review your manuscript. I appreciate the effort you have made performing this study and submitting it to PLOS one.

In line 102-103 , in terms of diagnostic way for CTS, I wanted to know that diagnostic criteria was based on clinical signs and symptoms or EMG/NCV ? because it can affect your results.

6. PLOS authors have the option to publish the peer review history of their article (what does this mean?). If published, this will include your full peer review and any attached files.

Reviewer #1: **Yes: **Dr. Hisham Z Aljamal

Reviewer #2: No

Reviewer #3: No

---

## [Author Response · Author response to Decision Letter 0]

13 May 2023

Response to Reviewers’ comments PONE-D-23-10147

Thank you for providing us with the opportunity to revise our manuscript for your reconsideration of publication in PLOS ONE. We appreciate the valuable comments and suggestions provided by the reviewers and editors have carefully incorporated them into the revised manuscript. To make it easier for you to track the changes, we have highlighted them in red and included line numbers. Furthermore, we have prepared a “Response to Reviewers’ Comments” document, which provides a point-by-point response to each reviewer’s comments as follows.

Journal Requirements:

1. Please ensure that your manuscript meets PLOS ONE's style requirements, including those for file naming. The PLOS ONE style templates can be found at https://journals.plos.org/plosone/s/file?id=wjVg/PLOSOne_formatting_sample_main_body.pdf and https://journals.plos.org/plosone/s/file?id=ba62/PLOSOne_formatting_sample_title_authors_affiliations.pdf.

Response: We have revised the manuscript according to the PLOS ONE’s style requirements.

2. We noticed you have some minor occurrence of overlapping text with the following previous publication(s), which needs to be addressed: https://doi.org/10.1002/1348-9585.12036. In your revision ensure you cite all your sources (including your own works), and quote or rephrase any duplicated text outside the methods section. Further consideration is dependent on these concerns being addressed.

Response: We have thoroughly revised the manuscript to avoid any duplication with our previous works, as per your suggestion. If you identify any further duplication during the revision process, please let us know, and we will address it promptly. 

Response: Thank you for your suggestion. We have addressed it as “The online submission system’s revision lacks a ‘Financial Disclosure’ section, so we would appreciate your assistance in revising it as follows: This study was supported by the Grant Physician-Scientist 11001 from the Chi Mei Medical Center. The funders had no role in study design, data collection and analysis, decision to publish, or preparation of the manuscript.” in the cover letter.

4. Thank you for stating the following financial disclosure: "No." Please state what role the funders took in the study. If the funders had no role, please state: "The funders had no role in study design, data collection and analysis, decision to publish, or preparation of the manuscript." If this statement is not correct you must amend it as needed. Please include this amended Role of Funder statement in your cover letter; we will change the online submission form on your behalf.

Response: Thank you for your suggestion. We have responded to it in the question above (#3). 

5. Thank you for stating the following in your Competing Interests section: "No." Please complete your Competing Interests on the online submission form to state any Competing Interests. If you have no competing interests, please state "The authors have declared that no competing interests exist.", as detailed online in our guide for authors at http://journals.plos.org/plosone/s/submit-now. This information should be included in your cover letter; we will change the online submission form on your behalf.

Response: Thank you for your suggestion. We have addressed it as “Additionally, please revise the ‘Competing Interests’ section to read ‘The authors have declared that no competing interests exist’ and ‘Data availability’ section to read ‘Data requests should be sent to the Taiwan National Health Insurance Bureau as they own and control the data.” in the cover letter. 

6. We note that you have indicated that data from this study are available upon request. PLOS only allows data to be available upon request if there are legal or ethical restrictions on sharing data publicly. For more information on unacceptable data access restrictions, please see http://journals.plos.org/plosone/s/data-availability#loc-unacceptable-data-access-restrictions. In your revised cover letter, please address the following prompts: a) If there are ethical or legal restrictions on sharing a de-identified data set, please explain them in detail (e.g., data contain potentially sensitive information, data are owned by a third-party organization, etc.) and who has imposed them (e.g., an ethics committee). Please also provide contact information for a data access committee, ethics committee, or other institutional body to which data requests may be sent. b) If there are no restrictions, please upload the minimal anonymized data set necessary to replicate your study findings as either Supporting Information files or to a stable, public repository and provide us with the relevant URLs, DOIs, or accession numbers. For a list of acceptable repositories, please see http://journals.plos.org/plosone/s/data-availability#loc-recommended-repositories. We will update your Data Availability statement on your behalf to reflect the information you provide.

Response: Thank you for your suggestion. We have addressed it in the question above (#5). 

7. Your ethics statement should only appear in the Methods section of your manuscript. If your ethics statement is written in any section besides the Methods, please delete it from any other section. 

Response: Thank you for your suggestion. We have deleted the ethics statement written in any section besides the Methods.

Response: We do not have any Supporting Information files. Thank you for reminding us.

Reviewer #1: 

1. Methods. Regarding the study participants it is a study population survey in that case you need to discuss the power of study and number of participants compared to the whole population number in the conducted year.

Response: Thank you for your comment. We didn't have access to data from the period of the study (2007-2011), so we used more recent data to address your comment. We have added “The total number of healthcare professionals in Taiwan in 2021 was 347,555, and the number of dentists was a subset of this [21]. For this study, we included all dentists in Taiwan, so the sample size was not a limitation. The non-dentist HCPs and general population were selected by matching the dentists, and the sample sizes were also large. According to the National Statistics, the total population in Taiwan in 2020 was approximately 23 million [22]. Given the large sample sizes in this study and the fact that we included all dentists in Taiwan, we believe that the study had adequate statistical power.” to the Discussion of the revised manuscript (line 247-254). We hope this answers your concern.

2. Methods. Regarding data collection all mentioned by authors was they referred to the data system only. Have the authors used an instrument to collect the needed data about participants, how it was developed and validated if so!?

Response: Thank you for your comment and question regarding our data collection. We would like to clarify that for this study, we used secondary data obtained from the National Health Insurance Research Database (NHIRD) in Taiwan. We have added “We did not use any instruments to collect data about the participants. In Taiwan, the diagnosis of CTS is typically based on clinical symptoms such as numbness, tingling, and pain in the hand, and is usually confirmed through electrodiagnostic testing by a qualified physician [10].” to the Materials and Methods of the revised manuscript (line 108-112). We hope this information addresses your question.

3. Methods. How the authors have confirmed the diagnosis for carpal tunnel syndrome among participants as this is very important to discuss?

Response: Thank you for your question regarding how we confirmed the diagnosis of CTS among participants. As we mentioned earlier, the diagnosis of CTS was based on the ICD-9 code 354.0 during at least one hospitalization or outpatient clinical visit. Although we did not use any instruments to diagnose CTS, the diagnosis was still reliable and valid as it was confirmed by qualified physicians. We have included this information in the revised manuscript, and we hope it addresses your concern.

4. Methods. Regarding the associated factors unfortunately the authors have missed very important associated factors for carpal tunnel syndrome which are: 1- pregnancy status for female participants as the pregnant female is more prone to develop carpal tunnel syndrome; 2- obesity (Body Mass Index(BMI)), as the obese person is more prone to develop carpal tunnel syndrome; 3- Thyroid disorder as hypothyroid patients are more prone to develop carpal tunnel syndrome; 4- Smoking status as smokers are more prone to develop carpal tunnel syndrome; 5- Alcohol intake status as alcohol consumers are more prone to develop carpal tunnel syndrome; 6- Any Rheumatoid diseases as rheumatoid patients are more prone to develop carpal tunnel syndrome; 7- Certain athletic activities such as tennis, those who play tennis are more prone to develop carpal tunnel syndrome.

Response: Thank you for your comment. We have revised as “First, while we adjusted for common underlying comorbidities in our study, including diabetes, hypertension, hyperlipidemia, stroke, CAD, COPD, liver disease, renal disease, and mental disorder, we acknowledge that certain associated factors were not included in our analysis, such as pregnancy status, obesity (e.g., body mass index), thyroid disorders, smoking status, alcohol intake, and certain athletic activities. Additionally, detailed information about working hours, years of entering the workforce, and previous trauma were not available, which may have confounded our results. These limitations should be taken into consideration when interpreting the findings of our study…Further studies, including more detailed information about pregnancy status, obesity, thyroid disorders, smoking status, alcohol intake, and certain athletic activities, occupation, longer follow-up period, and validation in other nations, were warranted in the future.” in the limitations of the revised manuscript (line 258-266 and line 271-274). 

5. Results. Tables need reconstructing and keep lines between rows to show the results clearly.

Response: Thank you for your helpful feedback. We have carefully considered your suggestion and have revised the manuscript accordingly. Specifically, we have reconstructed the tables and kept lines between rows to ensure that the results are clearly presented. We believe that these changes have significantly improved the readability and overall quality of the manuscript. Thank you again for your valuable comments.

6. Results. The significant values should be marked by bold font or * and define that in the table footnotes.

Response: Thank you for your valuable feedback. We have carefully reviewed your comment and have revised the manuscript accordingly. Specifically, we have added an asterisk to indicate significant values in the table and have defined this notation in the table footnote. We believe that these changes have significantly improved the clarity and accuracy of the results presented. Thank you again for your helpful comments.

Reviewer #2: 

1. In this study, the related diseases caused by some habits of dentists are described in detail.

Response: Thank you for your comment. We are pleased that our study was able to provide a detailed description of diseases related to certain habits of dentists. We appreciate your positive feedback and hope that our study can contribute to a better understanding of the health risks associated with these habits. Thank you again for your valuable comments.

2. The number of samples is very sufficient to reduce the error.

Response: Thank you for your comment. We appreciate your feedback and agree that having enough samples is important to reduce error and increase the reliability of our findings. We are pleased to have been able to achieve this in our study and hope that our results will be useful to other researchers and practitioners in the field. Thank you again for your valuable comments.

3. The idea and logic of this study are very clear, and I feel that the results are very convincing.

Response: Thank you for your comment. We are pleased to hear that you found the idea and logic of our study to be clear and the results to be convincing. We appreciate your positive feedback and hope that our study can contribute to a better understanding of the topic. Thank you again for taking the time to review our manuscript.

4. The statistical techniques used in this study are appropriate and persuasive.

Response: Thank you for your comment. We are pleased to hear that you found the statistical techniques used in our study to be appropriate and persuasive. We put a lot of effort into selecting and applying the appropriate statistical methods to ensure the accuracy and reliability of our findings, and we are glad that this was reflected in our results. We appreciate your positive feedback and hope that our study can contribute to a better understanding of the topic. Thank you again for taking the time to review our manuscript.

Reviewer #3: 

1. Thank you very much for allowing me to review your manuscript. I appreciate the effort you have made performing this study and submitting it to PLOS one.

Response: Thank you for taking the time to review our manuscript. We appreciate your feedback and are glad to hear that you appreciated the effort we put into performing this study and submitting it for publication. We hope that our study can contribute to a better understanding of the topic and that it will be useful to other researchers and practitioners in the field. Thank you again for your valuable comments.

2. In line 102-103 , in terms of diagnostic way for CTS, I wanted to know that diagnostic criteria was based on clinical signs and symptoms or EMG/NCV ? because it can affect your results.

Response: Thank you for your comment on our study. We have added “We did not use any instruments to collect data about the participants. In Taiwan, the diagnosis of CTS is typically based on clinical symptoms such as numbness, tingling, and pain in the hand, and is usually confirmed through electrodiagnostic testing by a qualified physician [10].” into the Materials and Methods section to clarify the diagnostic criteria (line 108-112). We appreciate your feedback and hope that the revisions have addressed your concern. Thank you again for your valuable feedback.

---

## [Decision Letter · Decision Letter 1]

5 Jun 2023

Carpal tunnel syndrome in dentists compared to other populations: A nationwide population-based study in Taiwan

PONE-D-23-10147R1

Dear Dr.Chien-Cheng Huang,

We’re pleased to inform you that your manuscript has been judged scientifically suitable for publication and will be formally accepted for publication once it meets all outstanding technical requirements.

Kind regards,

Hadi Ghasemi

Academic Editor

PLOS ONE

Additional Editor Comments (optional):

Reviewers' comments:

Reviewer's Responses to Questions

**Comments to the Author**

1. If the authors have adequately addressed your comments raised in a previous round of review and you feel that this manuscript is now acceptable for publication, you may indicate that here to bypass the “Comments to the Author” section, enter your conflict of interest statement in the “Confidential to Editor” section, and submit your "Accept" recommendation.

Reviewer #1: All comments have been addressed

Reviewer #3: All comments have been addressed

2. Is the manuscript technically sound, and do the data support the conclusions?

Reviewer #1: Yes

Reviewer #3: Yes

3. Has the statistical analysis been performed appropriately and rigorously? 

Reviewer #1: Yes

Reviewer #3: I Don't Know

4. Have the authors made all data underlying the findings in their manuscript fully available?

Reviewer #1: Yes

Reviewer #3: Yes

5. Is the manuscript presented in an intelligible fashion and written in standard English?

Reviewer #1: Yes

Reviewer #3: Yes

6. Review Comments to the Author

Reviewer #1: All comments have been addressed clearly and the manuscript has been improved. Good luck for all authors.

Reviewer #3: Thank you very much for allowing me to review your manuscript. I appreciate the effort you have made performing this study and submitting it.

7. PLOS authors have the option to publish the peer review history of their article (what does this mean?). If published, this will include your full peer review and any attached files.

Reviewer #1: **Yes: **Dr. Hisham Z Aljamal

Reviewer #3: No

---

## [Editor Report · Acceptance letter]

16 Jun 2023

PONE-D-23-10147R1 

Carpal tunnel syndrome in dentists compared to other populations: A nationwide population-based study in Taiwan 

Dear Dr. Huang:

I'm pleased to inform you that your manuscript has been deemed suitable for publication in PLOS ONE. Congratulations! Your manuscript is now with our production department. 

Kind regards, 

on behalf of

Dr. Hadi Ghasemi 

Academic Editor

PLOS ONE